# Testing of Rubber Composites Reinforced with Carbon Nanotubes

**DOI:** 10.3390/polym14153039

**Published:** 2022-07-27

**Authors:** Dana Bakošová, Alžbeta Bakošová

**Affiliations:** 1Department of Material Engineering, Faculty of Industrial Technologies in Púchov, Alexander Dubček University of Trenčín, Ivana Krasku 491/30, 020 01 Púchov, Slovakia; 2Department of Numerical Methods and Computational Modeling, Faculty of Industrial Technologies in Púchov, Alexander Dubček University of Trenčín, Ivana Krasku 491/30, 020 01 Púchov, Slovakia; alzbeta.bakosova@student.tnuni.sk

**Keywords:** nanocomposites, carbon nanotubes, mechanical properties, atomic force microscopy, dynamical mechanical analysis

## Abstract

Carbon nanotubes (CNTs) have attracted growing interest as a filler in rubber nanocomposites due to their mechanical and electrical properties. In this study, the mechanical properties of a NR/BR/IR/SBR compound reinforced with single-wall carbon nanotubes (SWCNTs) were investigated using atomic force microscopy (AFM), tensile tests, hardness tests, and a dynamical mechanical analysis (DMA). The tested materials differed in SWCNT content (1.00–2.00 phr) and were compared with a reference compound without the nanofiller. AFM was used to obtain the topography and spectroscopic curves based on which local elasticity was characterized. The results of the tensile and hardness tests showed a reinforcing effect of the SWCNTs. It was observed that an addition of 2.00 phr of the SWCNTs resulted in increases in tensile strength by 9.5%, Young’s modulus by 15.44%, and hardness by 11.18%, while the elongation at break decreased by 8.39% compared with the reference compound. The results of the temperature and frequency sweep DMA showed higher values of storage and loss moduli, as well as lower values of tangent of phase angle, with increasing SWCNT content.

## 1. Introduction

Polymer nanocomposites have attracted research interest and have found a broad field of application in recent years because of their enhanced material properties compared with original polymers. The extent of improvement depends, generally, on a number of parameters, including the type of nanofiller, particle size, aspect ratio, filler dispersion status, and surface properties, which determine the interaction between the filler and the polymer chain. Polymer carbon nanotube composites are objects of particular interest due to the structural characteristics of carbon nanotubes (CNTs) and their large surface area available for stress transmission, as well as their exceptionally high modulus of elasticity and excellent electrical and thermal properties [1].

Carbon nanotubes are seamless cylinders formed by rolling sheets of graphene atoms with open or closed ends. They can be divided into two main categories: single-wall carbon nanotubes (SWCNTs), with a diameter in the nanometer scale, and multi-wall carbon nanotubes (MWCNTs), consisting of several concentrically connected nanotubes or nanotubes rolled in a spiral. The diameter of a MWCNT can reach more than 100 nm. The length of CNTs can reach several micrometers or even millimeters. The structure of rolled CNTs is given by a chiral vector. Based on their structure, CNTs can be divided into the zigzag, chiral, and armchair types. The quality of CNTs relates to the fact that all carbons are bound in a hexagonal lattice except their ends. Similar to graphene, CNTs are chemically bound by sp2 bonds, which are extremely strong forms of molecular interaction. This property, combined with the natural tendency of carbon nanotubes to bond with van der Waals forces, provides an opportunity to develop ultra-high-strength, low-weight materials with high electrical and thermal conductivity. This makes them very attractive for many applications [2,3,4,5]. In addition to their electrical properties, which they inherit from graphene, CNTs also have unique thermal and mechanical properties. They have high Young’s modulus (up to 1 TPa) and tensile strength (11–63 GPa). They are very light with good thermal conductivity. As graphite, they have high chemical stability and are extremely resistant to corrosion, unless they are simultaneously exposed to high temperatures and oxygen [6].

Due to their properties, CNTs have been used as functional fillers in rubber compounds. Rubber nanocomposites reinforced with CNTs have been investigated in order to achieve required dynamic mechanical properties, gas resistance, flammability resistance, and thermal and electrical conductivity [7,8,9,10,11,12,13]. Several studies [14,15,16,17,18] have reported increases in tensile strength, elastic modulus, and hardness and decreases in the elongation at break of natural rubber composites reinforced with CNTs. In addition to improving the mechanical properties, the incorporation of conductive fillers into rubbers that are thermal and electrical insulators can produce composites with certain electrical conductivities. The potential applications of CNT-filled rubber composites vary from industrial applications such as rubber hoses, tire components, and sensing devices to electrically conductive systems and biomedical applications [19,20]. 

The final properties of CNT-reinforced elastomer nanocomposites mainly depend on the CNT type, the rate of CNT dispersion and their orientation in the matrix, the physical and chemical interactions of polymer chains with the CNTs, and the crosslinking chemistry of the rubbers. Orientation effects are mainly due to their high aspect ratio. The degree of alignment of nanotubes can be determined with X-ray diffraction and polarized Raman spectroscopy. As presented in [21], rolling direction affects the final alignment of the CNTs. If the composites are manufactured using extrusion or injection molding, tuning of the alignment degree can be achieved by regulating the shear rate, as well as the pressure applied. Other methods of alignment also include mechanical stretching, filtration, plasma-enhanced chemical vapor deposition, electrospinning, force-field-induced alignment, magnetic-field-induced alignment, liquid-crystalline-phase-induced alignment, etc. [22]. 

Carbon nanotubes are difficult to process due to their low dispersibility and their tendency to form aggregates [23], and in order to utilize CNT properties, several strategies have been proposed to improve the compatibility between polymer matrices and carbon nanotubes [24,25,26,27]. Acids and organic solvents are used to functionalize the carbon nanotubes [28,29,30], and another alternative is ionic liquids [31,32,33].

This study focuses on testing and evaluating selected material properties of rubber nanocomposites that differ in their contents of single-wall nanotubes and compares them with a rubber compound with the same base material but without nanofillers. The rubber compounds are examined using atomic force microscopy (AFM), hardness tests, tensile tests, and a dynamical mechanical analysis (DMA) in order to determine the influence of the CNT content on the mechanical properties. Microscopy is an essential tool for understanding the morphology of rubber compounds, including the size, shape and distribution of filler phases and particles, as well as for determining the effects of fillers and processing additives on the properties of rubber compounds. Atomic force microscopy (AFM) is a versatile and powerful analytical tool for the development and research of rubber materials. In [34,35,36,37,38], AFM was used to determine morphology, as well as for the observation of the microdispersion of the fillers in rubber compounds. AFM can also be used to characterize the local elasticity of rubber materials [39], to study the homogeneity of rubber compounds [40,41], to study the aging of rubber composites [42], and to determine the effects of fillers on the properties and qualities of compounds [42,43,44]. AFM force spectroscopy presents much information regarding the surface and mechanical properties of tested materials. Information on the elasticity and stiffness of individual macromolecules of the tested material can also be obtained from the measured curves. In [45,46], force spectroscopy was used to test rubber compounds. A more detailed, state-of-art review of advances in the use of AFM in polymer investigation was presented by Wang et al. in [47]. In the present study, AFM is used to observe the topography of the examined materials and to evaluate their elastic properties based on the measured spectroscopic curves.

## 2. Materials and Methods

### 2.1. Materials

The rubber compounds examined in this study had the same polymer matrices, and they differed in proportions of single-wall carbon nanotubes. These compounds were labelled as CNT 1–CNT 5, and they were compared with a reference compound without carbon nanofillers labelled as CNT 0. The composition of the base material is listed in Table 1, and the contents of single-wall carbon nanotubes of the individual compounds can be seen in Table 2. The used carbon nanotubes had a diameter of 2 nm, a length of 5–20 μm, and a purity of 95%.

The production process of the CNT 1–CNT 5 compounds was divided into several phases. The first phase involved the dispersion of the carbon nanotubes in a dispersant (ethanol) and a distillate aromatic extract (DAE), as nanotubes have tendency to form aggregates, which are difficult to process. This solution was heated to a temperature just above the boiling point of the solvent (80 °C) and then sonified for 120 min, using a mechanical ultrasonic sonifier with a probe capable of vibrating at appropriate ultrasonic frequencies (30 kHz) in order to induce the efficient dispersion of the nanotubes. As a part of the sonication, the dispersing agent also evaporated. The second phase involved dissolving the individual components of the rubber compound in an organic solvent (ethanol) with the addition of oil (DAE), followed by mixing at the temperature of 80 °C for 120 min until the components were mixed and part of the dispersant evaporated. Then the preparation process was followed with the phase of mixing the rubber compound with the carbon nanotube solution, which included a two-stage mixing process to mix the prepared solutions thoroughly and to ensure the evaporation of the excess solvent. Mixing was performed with a Farrel Technolab BR 1600 Banbury mixer. In the first stage, rubber components, including oil, carbon black, silica, resin, antioxidants, antidegradants, regenerates, stearin, wax, and nanotubes, were added in the oil solution. The temperature of the chamber, rotors, and cap was 70 °C. The total mixing time was 360 s. The maximum number of revolutions of the mixer was 55 rpm, and the maximum pressure was 196 kPa. The highest temperature of the compound was 150 °C. Before the second mixing stage, the compound was rolled for 30 min with a Servitec double roller in order to achieve better dispersion of the fillers in the compound. In the second stage of mixing, in order to vulcanize the compound, the remaining components were added: insoluble sulphur, zinc oxide (ZnO), and CBS. The compound was mixed for 150 s at a maximum temperature of 105 °C. Subsequently, the compound was rolled again with a double roller to achieve the required thickness of the compound for the preparation of test samples. The compound was then allowed to cool and stabilize in an oven at ambient temperature for 5 days. Test samples were prepared in accordance with the standards for individual tests of vulcanized, semi-finished products by cutting. 

### 2.2. Methods

#### 2.2.1. Tensile Test

Tensile tests give an orientation view about rubber material properties. Tensile curves are characteristic for rubber compounds. The following material characteristics were evaluated using the tensile test results: Tensile strength: the maximum tensile stress recorded during the elongation of the testing sample until the breaking moment;Elongation at break: tensile deformation of the sample working length in the breaking moment;Young’s modulus: defined as the initial slope of the stress–strain response.

Tensile tests were performed for the CNT 0–CNT 5 samples in accordance with the ISO 37 standard, which specifies a method for determining the tensile deformation characteristics of vulcanized and thermoplastic rubbers. Ten dumbbell-shaped samples of each compound made in accordance with the ISO 23529 standard were tested. The working length of the samples was 20 mm, and the loading speed was 100 mm/min.

#### 2.2.2. Hardness Test

Hardness is the ability of a material to withstand compressive forces. It depends on several factors, namely the Young’s modulus, the viscoelastic properties of the elastomer, the thickness of the test sample, the geometry of the indenter, the applied pressure, the rate of pressure increase, and the interval in which the hardness is recorded. Choosing the appropriate hardness test method is important in order to obtain accurate and reliable results. The most commonly used method for rubber compounds is the Shore A method, which follows the ISO 7619-1 standard, according to which a test material with a thickness of 6 mm is required. 

Hardness measurements can be used to roughly estimate the Young’s modulus. The best-known correlation of hardness values to Young’s modulus was introduced by A.N. Gent [48] in 1958 and is given by the following equation:(1)$$E=\frac{0.0981\left(56+7.62336S\right)}{0.137505\left(254-2.54S\right)}\;\left(\mathrm{MPa}\right),\;\;$$
where *E* is Young’s modulus, and *S* is Shore A hardness in the range of 20–80 Shore A. There are several other correlations, such as the equation postulated by Ruess [49,50]:(2)$$log_{10}E=0.0235S-0.6403,$$
where *E* (MPa) is Young’s modulus, and *S* is Shore A hardness. The correlation by Lindeman [51] is another example:(3)$$E=11.427S-0.4445S^{2}+0.0071S^{3}\;\left(psi\right),$$
where *E* (psi) is Young’s modulus, and *S* is Shore A hardness. 

In this study, the correlations mentioned above were used to estimate Young’s modulus, and the estimated values were compered to results from the tensile tests.

#### 2.2.3. Dynamical Mechanical Analysis

Rubber compounds are viscoelastic materials, and a dynamic mechanical analysis (DMA) is often used to characterize their viscoelastic properties. In this study, a Pyris Diamond DMA analyzer was used for the DMA of the CNT 0–CNT 5 rubber compounds. Samples were prepared in the form of a strip with measurements of 20 mm × 10 mm × 2.1 mm. They were subjected to tensile loading in the temperature range of −80–100 °C using a heating rate of 5 °C/min at a frequency of 1 Hz. A cryogenic nitrogen vessel was used to achieve low temperatures. The samples were also subjected to frequencies of 0.01 Hz, 0.05 Hz, 0.2 Hz, 0.5 Hz, 1 Hz, 5 Hz, 10 Hz, 20 Hz, and 50 Hz at 20 °C. The frequency and temperature dependencies of the storage modulus *E*′, the loss modulus *E**″*, and the tangent of the phase angle *tan δ* were evaluated. 

#### 2.2.4. Atomic Force Microscopy

Atomic force microscopy (AFM) is a nonoptical imaging technique that allows accurate and nondestructive measurements of topographic, mechanical, electrical, magnetic, chemical, and optical properties of a sample surface at very high resolutions. An AFM microscope works on the principle of measuring the intermolecular force. It uses a cantilever with an attached, sharp, several-micrometers-long tip, which scans the surface of the sample. Atomic forces between the tip and the sample surface result in the bending of the cantilever. To detect any changes in the cantilever deflection, a laser beam is used. It is directed at the end of the cantilever, from which it is reflected into a photodetector. Change in the distance of the tip from the surface causes a change in the force and bending of the cantilever and, thus, the direction of the reflection of the laser beam into the photodetector changes. The deflection of the laser beam is recorded, and the resulting surface topography is generated with further software processing [52]. 

In spectroscopic measurements, the deflection of the cantilever tip is recorded as a function of the force and distance between the AFM probe and the sample. Due to the different stiffnesses of the tested systems, the stiffness constant of the AFM probe must be higher than the stiffness of the examined sample [37,53,54]. The spectroscopic curves were sufficiently specific for each material; however, they could be divided into general characteristic sections, as shown in Figure 1. The solid line shows the curves measured in a vacuum. The dashed line is the variation of the curves measured in air with the presence of layers of moisture and microscopic impurities. 

Between the A and B points, the tip and the sample are far apart, and there is no deflection of the cantilever with the tip. At the B point, long-distance interactions, mainly of Van der Waals and electrostatic origins, occur upon approach. At the C point, the tip touches the surface of the sample. The shape of the curve is also influenced by the surface moisture and impurities. The C–D section is characterized by further approach of the tip to the sample while they are physically in contact, and it results in pressing the tip against the sample surface and the deflection of the probe. At the D point, the sample surface is punctured due to the maximum force that the sample surface is able to withstand. The D point characterizes the end of the approach and the beginning of the departure of the tip from the surface. According to the slope of the C–D section, the Young’s modulus of the probe–surface system can be evaluated. If the probe is softer than the sample surface, the slope of the curve mostly presents the modulus of the probe; otherwise, if the stiffness of the probe is higher, the slope of the section allows the Young’s modulus of the sample to be examined. If the C–D and D–E sections are not parallel, the time-reversible elastic or plastic deformation of the sample can be evaluated. The probe deflection is neutral at the E point. The probe moves away from the surface between the E and F points, and it begins to tilt towards the sample due to attractive or adhesive forces. In a vacuum, Van der Waals and electrostatic forces act on the tip, and in an air environment, the tip is also subjected to capillary force from the moisture on the surface. The F point is a separation point at which the maximum adhesive force acts between the tip and the surface of the sample, and it gives information for adhesion evaluation. The number of the separation points depends on the viscosity and thickness of the surface layers (moisture, impurities, and grease). The probe separates from the surface at the G point after overcoming the adhesive force [53,55].

General approximation and Snedonn’s model [56,57] were used to evaluate the measured data and to calculate the ratios of the Young’s moduli. Snedonn’s model formulates the dependence between Young’s modulus *E* and the load gradient *dP/dh* and is given by the following equation:(4)$$\frac{dP}{dh}=\frac{2A^{1/2}}{\pi^{1/2}}E\;\left[Nm^{-1}\right],\;$$
where *A* (m^2^) is the contact area, and *E* (Pa) is a combined modulus of the elasticity of the probe and the examined surface, which is given by the following equation:(5)$$E={\left\{\left[\left(1-\nu_{m}^{2}\right)/E_{m}\right]+\left[\left(1-\nu_{c}^{2}\right)/E_{c}\right]\right\}}^{-1}\;\left[Pa\right],$$
where *E_m_*, *E_c_* (Pa) are the Young’s moduli of the examined material and the cantilever, respectively; *ν_m_*, *ν_c_* (-) are the Poisson ratios of the examined material and the cantilever, respectively, *h* (m) is the indentation depth, and *P* (N) is the normal load. It can be assumed that the Young’s modulus of the cantilever is much higher compared with that of the examined material (*E_c_* ≫ *E_m_*). Therefore, Equation (5) can be simplified to *E_m_ = E*, and the following equations representing the moduli of two different samples *E*1 and *E*2 in (6) and (7), respectively, and their ratio (8) can be derived:(6)$$\frac{dP_{1}}{dh_{1}}=\frac{2A^{1/2}}{\pi^{1/2}}E_{1}\Rightarrow E_{1}=\frac{dP_{1}}{dh_{1}}\frac{\pi^{1/2}}{2A^{1/2}},$$
(7)$$\frac{dP_{2}}{dh_{2}}=\frac{2A^{1/2}}{\pi^{1/2}}E_{2}\Rightarrow E_{2}=\frac{dP_{2}}{dh_{2}}\frac{\pi^{1/2}}{2A^{1/2}},$$
(8)$$\frac{E_{1}}{E_{2}}=\frac{\frac{dP_{1}}{dh_{1}}\;\frac{\pi^{1/2}}{2A^{1/2}}}{\frac{dP_{2}}{dh_{2}}\;\frac{\pi^{1/2}}{2A^{1/2}}}\Rightarrow \frac{E_{1}}{E_{2}}=\frac{\frac{dP_{1}}{dh_{1}}}{\frac{dP_{2}}{dh_{2}}}.$$

After a linear approximation (9) of the C–D section of the spectroscopic curve from which the Young’s modulus can be determined in order to find the slope *k*, Equation (11), which expresses the ratio of the Young’s moduli of the two samples, can be formulated:(9)$$y=kx+q,\;\mathrm{where}\;k=\frac{dP}{dh},$$
(10)$$\frac{dP_{1}}{dh_{1}}=k_{1}\;\mathrm{and}\;\frac{dP_{2}}{dh_{2}}=k_{2},$$
(11)$$\Rightarrow \frac{E_{1}}{E_{2}}=\frac{k_{1}\;}{k_{2}}\to E_{2}=\frac{k_{2}E_{1}}{k_{1}}.$$

## 3. Results

### 3.1. Tensile Test Results

In relation to the tensile test, each CNT 0–CNT5 compound was subjected to ten measurements, and subsequently, the tensile strength, elongation at break, and Young’s modulus were determined. The average values of these material properties are listed in Table 3, and a graphical comparison of the examined rubber compound properties can be seen in Figure 2. 

The tensile strength and Young´s modulus of the compounds with nanofillers in the form of the carbon nanotubes (CNT 1–CNT 5) increased with increasing content in the nanofiller. The tensile strength of the CNT 5 compound with the highest nanofiller content (2.00 phr) was higher by 9.5% compared with CNT 0 (without nanofillers). The Young´s modulus of CNT 5 was higher by 15.44% compared with CNT 0. The presence of SWCNTs in the tested compounds caused reductions in the values of the elongation at break. The elongation at break of CNT 5 decreased by 8.39% compared with CNT 0.

### 3.2. Hardenss Test Results

Approximately ten Shore A hardness measurements for each compound were performed, and Young´s moduli were calculated using Equations (1)–(3). The results are listed in Table 4, and dependence of Young’s modulus on hardness given by Equations (1)–(3) is shown in Figure 3.

The hardness of the compounds increased with increasing content in nanotubes. The hardness of CNT 5 was higher by 11.18% compared with the CNT 0 compound without the nanotubes. 

### 3.3. Evaluation of Rubber Compounds Using Atomic Force Microscopy

An NT-206 atomic force microscope was used to evaluate the rubber compounds, and along with the appropriate hardware and software, it allowed the analysis of the topography and micromechanical properties of the solid surfaces up to a nanometer-level resolution. The topography examples of the examined compounds are shown in Figure 4. The spectroscopic curves were measured for ten different locations of each compound. The C–D section of the spectroscopic curve (Figure 1) was approximated using a linear function in order to evaluate the Young´s moduli of the individual compounds. The ratios of the Young’s moduli of the compounds with nanotubes (CNT 1–CNT 5) to the compound without carbon nanotubes (CNT 0) were determined. The examples of spectroscopic curves for the CNT 0 and CNT 5 compounds are shown in Figure 5, and the slopes of the linear functions approximating the C–D section of the spectroscopic curves are listed in Table 5. 

Based on Equation (11), the values of the Young’s moduli of the CNT 1–CNT 5 nanocomposites were determined with respect to the CNT 0 reference compound without carbon nanotubes, and they are listed in Table 6. These ratios were calculated using the average slope values (Table 5).

The slope values of the spectroscopic curves of the individual compounds obtained from the linear approximation of the C–D section differed slightly, which indicated a slight inhomogeneity. By comparing the average values of the slopes, it can be stated that the CNT 5 compound had the highest value for the Young’s modulus, and the CNT 0 compound had the lowest value of the modulus. 

### 3.4. Dynamical Mechanical Analysis Results

The DMA was performed, during which the samples were subjected to a tensile loading in the temperature range of −80–100 °C and to the frequencies of 0.01 Hz, 0.05 Hz, 0.2 Hz, 0.5 Hz, 1 Hz, 5 Hz, 10 Hz, 20 Hz, and 50 Hz. The temperature dependencies of *E′, E″*, and *tan δ* at a frequency of 1 Hz for the CNT 0–CNT 5 rubber compounds can be seen in Figure 6 and Figure 7. The frequency dependencies of *E′, E″*, and *tan δ* at the temperature of 20 °C are shown in Figure 8.

The dependence of the elastic portion of the complex elasticity modulus on temperature (−80–100 °C) at a frequency of 1 Hz is shown in Figure 6. The storage modulus *E*′ of the tested rubber compounds increased with increasing CNT proportion. The storage modulus reflects the elastic properties of the tested materials and the renewable energy in the deformed samples. At a low temperature, the modulus *E*′ had a relatively high value that was attributed to the inert semicrystalline structure, and as the mobility of the polymer chains increased with temperature, the elastic modulus decreased. 

The dependence of the viscous portion of the complex elasticity modulus on temperature (−80–100 °C) at a frequency of 1 Hz can be seen in Figure 7a. The loss modulus *E″* corresponds to the viscous properties of a viscoelastic material and is a measure of a material’s ability to dissipate energy in the form of heat due to viscous movements in the material. The values of the loss modulus were significantly lower than the values of the storage modulus, with elastic properties predominant in the compounds. The loss modulus increased slightly with the increasing CNT content in the tested compounds. 

Figure 7b shows the dependence of the tangent of phase angle *tan δ* on temperature at a frequency of 1 Hz and a temperature range of −80–100 °C. The *tan δ* is determined from the ratio of the loss modulus to the storage modulus and represents the ratio of dissipated, lost energy to the energy stored during the deformation cycle. The *tan δ* characterized the damping material properties, which decreased with the increased CNT content and could be attributed to CNT stiffness. The glass transition temperature *T_g_* determined from the peak of the temperature dependency of *tan δ* was around −50 °C, and there was not noticeable change in the *T_g_* after addition of nanotubes.

The dependencies of the storage modulus *E′*, the loss modulus *E″* and the *tan δ* on the frequency at 20 °C can be seen in Figure 8. The storage modulus *E′*, the loss modulus *E″*, and the *tan δ* showed increasing tendency for all the compounds within the investigated frequency interval. With further increase in the frequency, increases in the storage moduli and decreases in the loss moduli and *tan δ* past their peaks were expected due to the viscoelastic nature of rubber compounds. The CNT 5 nanocomposite with the highest SWCNT content showed higher *E′* and *E″* values and lower values of *tan δ* compared with CNT 0.

## 4. Discussion

The tensile test results showed a reinforcing effect of the single-wall carbon nanotube filler. As can be seen in Table 3 and Figure 2, the tensile strength and Young’s modulus values increased with the increasing SWCNT content, and the elongation at break values decreased. The same trends were observed in [16,17,18,19,20]. The increase in the tensile strength and the Young’s modulus could be attributed to good dispersion and interatomic interaction between the rubber matrix and the nanofiller, as the CNTs (with their high aspect ratio) could improve the crosslinking of the compound [58]. The strengthening effect of CNTs was also reflected in the hardness of the tested compounds, and resistance to the penetration of foreign objects into the material increased. Hardness can also be used to roughly estimate Young’s modulus with a suitable correlation model. The estimation of Young’s modulus based on Shore A hardness measurements (calculated using Equations (1)–(3)) and its comparison to the tensile test results can be seen in Figure 9. In comparison to the tensile tests results, the closest estimation of Young’s modulus was calculated using Lindeman’s Equation (3). However, such calculation of Young’s modulus provides just approximate values and more measurements are needed to determine the most appropriate correlation model for this type of nanocomposite. The advantages of Young’s modulus estimation from hardness measurements are that it is quick and inexpensive, and it can find application, for example, in the testing of material properties during a production process [59].

AFM spectroscopic curves can be a good tool for monitoring the Young’s moduli of various materials, as they provide opportunities to compare materials in terms of their stiffness and elastic behavior. If the value of the Young’s modulus of the reference material is known, it allows the calculation of values of the moduli of other materials. Based on the results of force spectroscopy, the Young’s modulus values of the CNT 1–CNT 5 nanocomposites were calculated using Equation (11), as well as the reference value of the Young’s modulus (E_CNT 0_ = 3.032 MPa) from the tensile test. The results obtained from the AFM and spectroscopic curves were comparable to the results obtained from the static tensile test and are summarized in Table 7 and graphically represented in Figure 10. Force spectroscopy also allows the comparison of the slope of a spectroscopic curve and the local moduli within one sample in order to determine the properties of selected material phases and to evaluate its homogeneity. With the results summarized in the table, it is possible to observe a larger variance of local values compared with the reference sample. This could be caused by the presence of CNTs in the polymer matrix, as number of studies have suggested that the interaction of a polymer matrix with CNTs results in an interfacial region with properties and a morphology different than the bulk [60,61]. Using AFM, the significant agglomerates of the CNTs were not detected. However, for a thorough characterization of the dispersion, further research is needed using an AFM microscope with better resolution or with different methods, such as scanning electron microscopy [62] or transmission electron microscopy [63]. These might be useful for determining the efficiency of the mixing process, as well as for potentially further improving the material characteristics of the tested materials.

The viscoelastic properties of a rubber composite depend on the interactions of its components, the crystalline behavior, and the extent of crosslinking between the polymer chains and the filler. These properties improve with the addition of suitable fillers [58]. A temperature and frequency sweep DMA was performed to investigate the viscoelastic properties of the tested nanocomposites. The values of the storage modulus and the loss modulus increased, and the tangent of phase angle decreased with the increasing content in SWCNTs. Similar trends have been observed in [30,64,65,66], where different types of rubber nanocomposites filled with SWCNTs have been investigated. Dynamic stiffness improved with the addition of SWCNT nanofiller.

## 5. Conclusions 

In recent years, rubber nanocomposites reinforced with a low-volume fraction of carbon-based nanofillers have attracted research interest due to their properties. The incorporation of nanofillers into various elastomers has been found to improve their overall mechanical properties. The properties of rubber nanocomposites depend significantly on the structure of the polymer matrices, the nature of the nanofillers, and the technological processes used for their preparation. A uniform dispersion of a nanofiller in a rubber matrix is a general prerequisite for achieving the desired material characteristics. 

In the present study, the influence of single-wall CNTs on the mechanical properties of a NR/BR/IR/SBR compound were investigated. Five compounds that differed in SWCNT content (1.00–2.00 phr) and one without CNTs were tested and compared mutually. It was observed that the tensile strength, the Young’s modulus, and the hardness increased with increasing SWCNT content, while the elongation at break decreased. A comparison of the DMA results for the compounds showed an increase in the loss modulus and the storage modulus and a decrease in the tangent of phase angle values with increasing CNT content. The addition of a small amount of well-dispersed nanotubes in the rubber compound improved its mechanical properties. AFM force spectroscopy combined with Snedonn’s model was used to evaluate the local elasticity of the samples, and the results showed good agreement with the tensile test results. AFM force spectroscopy was a useful tool for obtaining information about the elasticity and stiffness of individual phases of the tested material. 

## Figures and Tables

**Figure 1 polymers-14-03039-f001:**
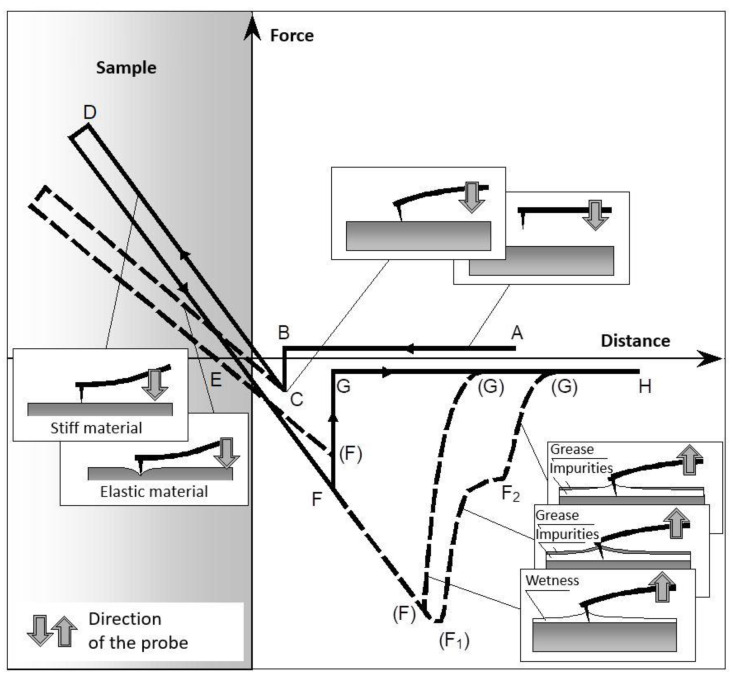
Schematic of force spectroscopic curve [55].

**Figure 2 polymers-14-03039-f002:**
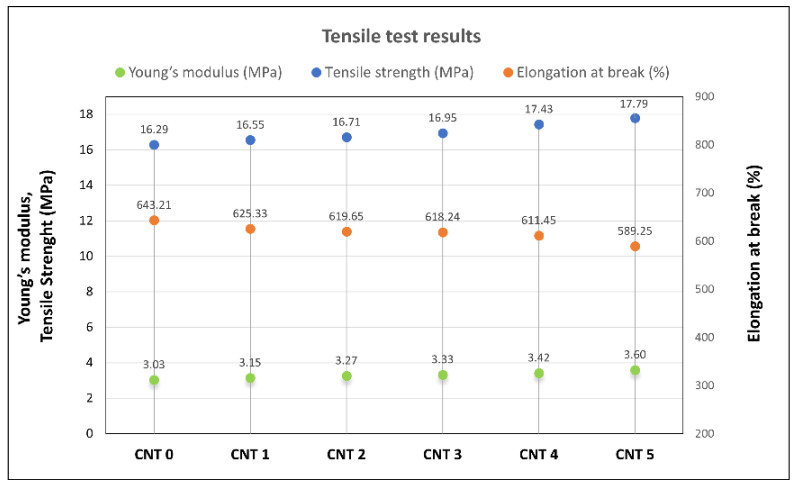
Tensile test results.

**Figure 3 polymers-14-03039-f003:**
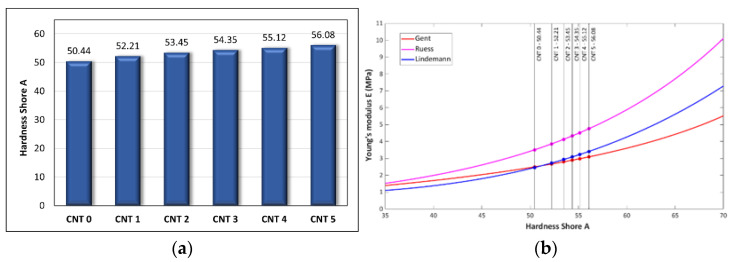
(**a**) Hardness test results and (**b**) estimation of Young’s modulus from Shore A hardness.

**Figure 4 polymers-14-03039-f004:**
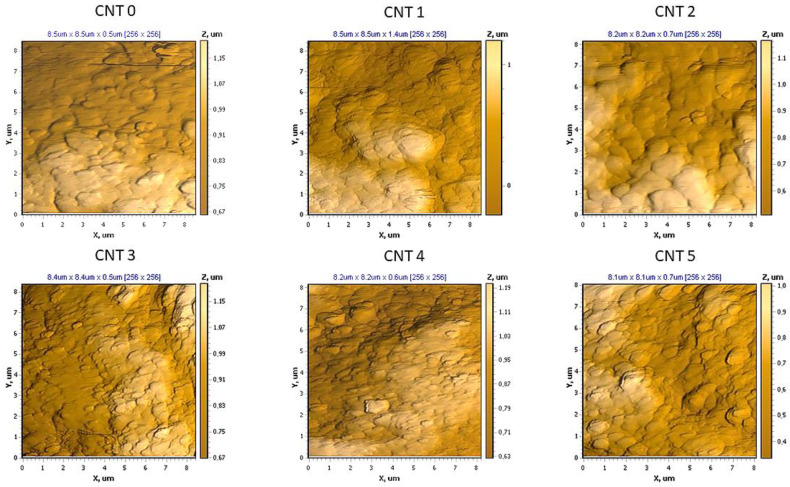
Examples of the topography of the individual compounds obtained by AFM.

**Figure 5 polymers-14-03039-f005:**
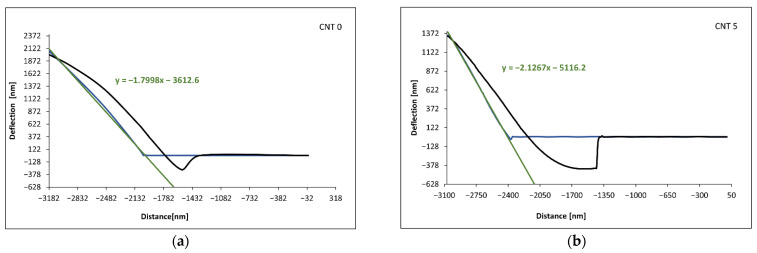
Examples of the spectroscopic curves of the compounds: (**a**) CNT 0 and (**b**) CNT 5.

**Figure 6 polymers-14-03039-f006:**
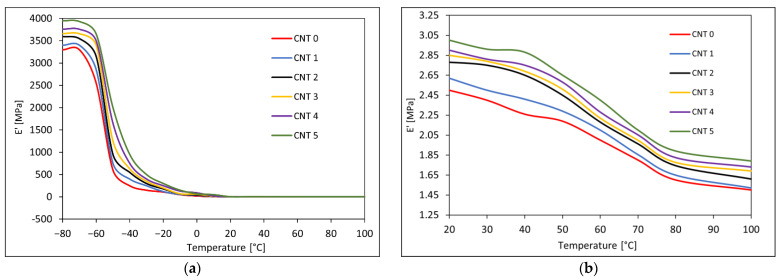
(**a**) Temperature dependence of storage modulus *E′* at frequency of 1 Hz and temperature range of −80–100 °C; (**b**) detail of temperature dependence of storage modulus *E′* at frequency of 1 Hz and temperature range of 20–100 °C.

**Figure 7 polymers-14-03039-f007:**
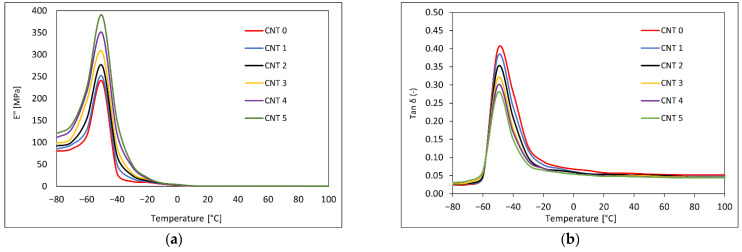
(**a**) Temperature dependence of loss modulus *E″* at frequency of 1 Hz and temperature range of −80–100 °C; (**b**) temperature dependence of the tangent of phase angle *tan*
*δ* at frequency of 1 Hz and temperature range of −80–100 °C.

**Figure 8 polymers-14-03039-f008:**
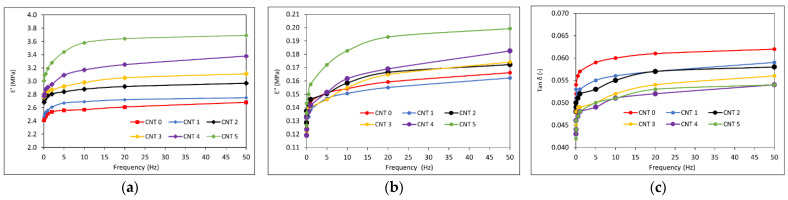
Frequency dependencies (frequency range: 0.1–50 Hz) at a temperature of 20 °C: (**a**) the storage modulus *E′*, (**b**) the loss modulus *E*″, and (**c**) the tangent of phase angle *tan*
*δ*.

**Figure 9 polymers-14-03039-f009:**
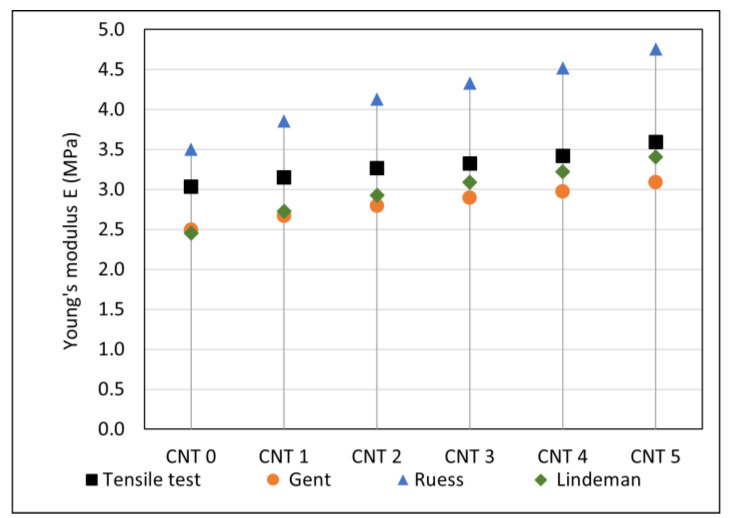
Comparison of Young’s modulus from tensile tests with values estimated from hardness tests.

**Figure 10 polymers-14-03039-f010:**
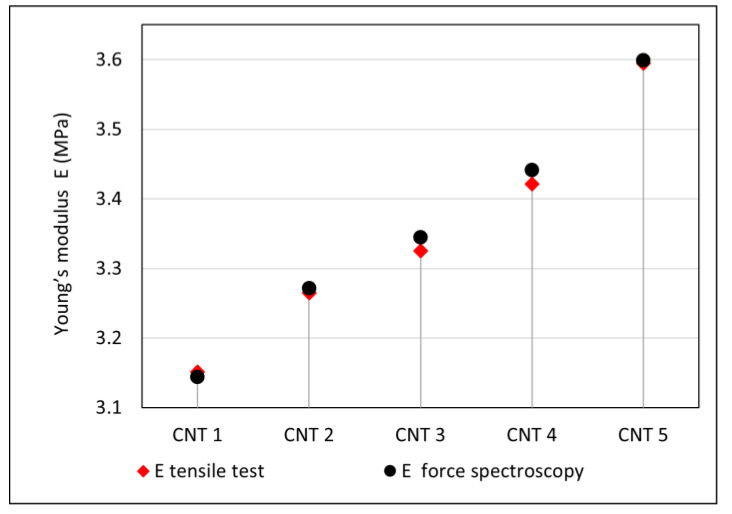
Comparison of Young’s modulus obtained from tensile test and force spectroscopy.

**Table 1 polymers-14-03039-t001:** Composition of the examined compounds.

Component	Amount (phr)
natural rubber (NR)	40
butadiene rubber (CIS BR)	20
isoprene rubber (CIS IR)	10
styrene-butadiene rubber (SBR 1500)	30
carbon black filler (N 339)	30
reclaimed rubber	10
silica filler ULTRASIL	8
resin	2
antidegradant	3
antioxidant	2
vulcanization activator—STEARIN	2
distillate aromatic extract DAE	2
oxidized polyethylene wax (OPW)	2
vulcanization activator—ZnO	2.5
vulcanizing agent—insoluble sulphur 67%	3.3
sulfenamide vulcanization accelerator—CBS	1.1

**Table 2 polymers-14-03039-t002:** The contents of the single-wall carbon nanotubes in the compounds.

Compound label:	CNT 0	CNT 1	CNT 2	CNT 3	CNT 4	CNT 5
**Content of carbon nanotubes (phr):**	0.00	1.00	1.25	1.50	1.75	2.00

**Table 3 polymers-14-03039-t003:** Tensile test results.

Compound	Tensile Strength (MPa)	Elongation at Break (%)	Young’s Modulus (MPa)
CNT 0	16.29 ± 0.35	643.21 ± 13.21	3.03 ± 0.03
CNT 1	16.55 ± 0.27	625.33 ± 12.55	3.15 ± 0.02
CNT 2	16.71 ± 0.31	619.65 ± 14.25	3.27 ± 0.03
CNT 3	16.95 ± 0.39	618.24 ± 10.22	3.33 ± 0.04
CNT 4	17.43 ± 0.32	611.45 ± 11.45	3.42 ± 0.04
CNT 5	17.79 ± 0.41	589.25 ± 13.99	3.60 ± 0.04

**Table 4 polymers-14-03039-t004:** Hardness test results and estimation of Young´s modulus using Equations (1)–(3).

Compound	Shore A Hardness	Young’s Modulus (MPa)		
		Gent’s Equation	Ruess’s Equation	Lindeman’s Equation
CNT 0	50.44 ± 0.21	2.50 ± 0.35	3.51 ± 0.43	2.46 ± 0.36
CNT 1	52.21 ± 0.35	2.67± 0.43	3.86 ± 0.54	2.73 ± 0.45
CNT 2	53.45 ± 0.39	2.80 ± 0.49	4.13 ± 0.57	2.93 ± 0.50
CNT 3	54.35 ± 0.24	2.89 ± 0.37	4.33 ± 0. 45	3.09 ± 0.39
CNT 4	55.12 ± 0.29	2.98 ± 0.38	4.52 ± 0.47	3.23 ± 0.40
CNT 5	56.08 ± 0.32	3.09 ± 0.41	4.76 ± 0.52	3.41 ± 0.44

**Table 5 polymers-14-03039-t005:** The slopes of the linear functions approximating the spectroscopic curves.

Measurement	CNT 0	CNT 1	CNT 2	CNT 3	CNT 4	CNT 5
**1**	−1.7998	−1.8642	−1.9354	−1.9824	−2.0322	−2.1267
**2**	−1.7954	−1.8524	−1.9521	−1.9724	−2.0452	−2.1454
**3**	−1.7969	−1.8852	−1.9754	−1.9853	−2.0563	−2.1541
**4**	−1.7912	−1.8921	−1.9551	−1.9621	−2.0725	−2.1145
**5**	−1.7954	−1.8245	−1.9254	−1.9994	−2.0168	−2.1354
**6**	−1.7945	−1.8526	−1.9278	−1.9685	−2.0698	−2.1078
**7**	−1.7997	−1.8354	−1.9245	−1.9824	−2.0597	−2.1298
**8**	−1.7921	−1.8759	−1.9154	−1.9974	−2.0125	−2.1758
**9**	−1.7991	−1.8875	−1.9285	−1.9899	−2.0137	−2.1267
**10**	−1.7991	−1.8522	−1.9354	−1.9678	−2.0045	−2.1045
**Avarage value of slope k_CNT i_**	**−1.7963 ± 0.0010**	**−1.8622 ± 0.0072**	**−1.9375 ± 0.0057**	**−1.9808 ± 0.0040**	**−2.0383 ± 0.0081**	**−2.1321 ± 0.0069**

**Table 6 polymers-14-03039-t006:** Young’s modulus ratios of compounds CNT 1–CNT 5 with respect to CNT 0.

Rubber Compound	Young’s Modulus Ratios $E_{CNT\;i}=\frac{k_{CNT\;i\;}E_{CNT0}}{k_{CNT\;0}}$
CNT 1	*E_CNT_*_1 =_ 1.037*E_CNT_*_0_
CNT 2	*E_CNT_*_2 =_ 1.079*E_CNT_*_0_
CNT 3	*E_CNT_*_3 =_ 1.103*E_CNT_*_0_
CNT 4	$E_{CNT4}=1.135E_{CNT0}$
CNT 5	$E_{CNT5}=1.187E_{CNT0}$

**Table 7 polymers-14-03039-t007:** Comparison of Young’s modulus obtained from tensile tests and AFM force spectroscopy.

Compound	Young’s Modulus (MPa)	
	AFM	Tensile Test
CNT 1	3.144	3.151
CNT 2	3.272	3.265
CNT 3	3.344	3.325
CNT 4	3.441	3.421
CNT 5	3.599	3.595

## Data Availability

Data are contained within the article.

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
