# Peer review of "Testing of Rubber Composites Reinforced with Carbon Nanotubes"

_polymers, 2022, doi:10.3390/polym14153039_

Round 1

Reviewer 1 Report

The paper may be published by Polymers after some revision:

 1 – The equation presented in Table 6 needs to be corrected.

2 – A relevant point is that it would be interesting if the authors enlighten us with a clearer explanation of how the rubber compounds were prepared. What is a special oil solution or a suitable organic solvent?

It was not clear how the mixture between CNT and the rubbers was performed. There is no information about the drying stage of the solutions. How and where were the compounds mixed? What were the vulcanization conditions?

3 - The compounds did not show a significant improvement in the mechanical properties (except for hardness) with the CNT addition since we are talking about nanofillers. So, was a good dispersion really obtained? How was aggregation avoided?

4 – In “3.4. Dynamical Mechanical Analysis Results” section, Figures 6 to 8 and Figures 10 to 12 could be grouped together, as well as the discussion of the results, in order to compose a more concise and less repetitive text.

5 – In addition, one of the highlights of the paper is the use of AFM to determine the Young's rubber compound moduli. This aspect could be more valued in the conclusion and abstract. The abstract could be improved, with the addition of some numerical data.

Reviewer 2 Report

Manuscript title: Testing of Rubber Composites Reinforced with Carbon Nanotubes

Manuscript ID: polymers-1809482

Review comments: The work is interesting. The manuscript describes about mechanical properties of a NR/BR/IR/SBR compound reinforced with single-wall carbon nanotubes (SWCNTs) using atomic forced microscopy (AFM), tensile tests, hardness tests and dynamical mechanical analysis (DMA). It is very good for the readers to this journal. But it needs the following corrections before accepting its final publication.

1.    In the abstract re-written the whole part using one tense. Mixing is not correct.

2.    In Table 3, use same decimal for all columns. 3rd columns use 2 digits after decimal point and 2nd & 4th columns are different.

3.     In Fig. 6 & 7 can be combined for better presentation numbering a & b like Fig.5. 

4.      In Fig. 8 & 9 can be combined for better presentation numbering a & b. 

5.    In Fig. 10, 11 & 12 can be combined for better presentation numbering a,b,c. 

6.    The English is also need to edit moderately.

Reviewer 3 Report

In this contribution, the authors investigated the reinforcing effects of carbon nanotubes (CNT) in rubber composites. The increasing CNT contents improve the tensile strength, Young’s modulus and hardness of rubber composites, and reduce the elongation at break. The thorough characterizations and convincing results support the authors’ conclusions, and this research is inspiring for the readership of Polymers. Therefore, I would recommend its publication if the following questions and comments are addressed.

1. The CNT employed in this work has a diameter of 2 nm (Line 99), claimed as sing-wall carbon nanotubes (SWCNT). However, the introduction categorizes SWCNT with a diameter of less than 1 nm (Line 39). Can authors justify this controversy?

2. The blending process utilizes various solvents, including a special oil solution for CNT, and suitable organic solvents for rubber composites, etc.. After the blending, are there residual solvents in the CNT-filled rubber? Are the blending temperatures, such as 150 °C, sufficient to evaporate those solvents? If not, do the residual solvents affect the homogeneity and mechanical performance of CNT-filled rubber?

3. Given the high aspect ratio of CNT, does the mixing induce parallel orientation of CNT in the shearing direction? In addition, can the dispersion of CNT be characterized?

4. In Figure 5a and b, the equations do not fit the C-D section of spectroscopic curves. For example, in Figure 5a, the green fitting line of C-D section goes through two coordinates (-3480, 1872) and (-2240, -628), giving a linear function of y = -2.016x - 5144.

5. Considering the complex composition of the rubber matrix in this study, what are the specific interactions between the CNT and rubber matrix? 
